# Nutritional Ergogenic Aids in Cycling: A Systematic Review

**DOI:** 10.3390/nu16111768

**Published:** 2024-06-05

**Authors:** Alberto Valiño-Marques, Alexandre Lamas, José M. Miranda, Alberto Cepeda, Patricia Regal

**Affiliations:** 1Faculty of Sport Sciences, European University of Madrid, 28670 Madrid, Spain; 22304557@live.uem.es; 2Department of Analytical Chemistry, Nutrition and Bromatology, University of Santiago de Compostela, 27002 Lugo, Spain; alexandre.lamas@usc.es (A.L.); alberto.cepeda@usc.es (A.C.); patricia.regal@usc.es (P.R.)

**Keywords:** cycling, ergogenic aid, dietary supplements, athletic performance, endurance training, synergistic effect, Australian Institute of Sport (AIS)

## Abstract

This systematic review aimed to evaluate the effectiveness of the independent or combined use of nutritional ergogenic aids belonging to Group A of the ABCD classification by the Australian Institute of Sport (AIS) in the context of cycling (caffeine, creatine, sodium bicarbonate, beta-alanine, nitrates, and glycerol). A comprehensive search was carried out using three databases: PubMed, Scopus, and Web of Science. All the databases were searched for Randomized Controlled Trials or crossover design studies assessing the effects of supplementation on cycling performance in comparison with placebos in healthy adults. The methodological quality of each study was evaluated using the Physiotherapy Evidence Database scale. Thirty-six articles involving 701 participants were included in this review, examining supplementation with caffeine (n = 5), creatine (n = 2), sodium bicarbonate (n = 6), beta-alanine (n = 3), and nitrates (n = 8). Additionally, supplemental combinations of caffeine and creatine (n = 3), caffeine and sodium bicarbonate (n = 3), caffeine and nitrates (n = 1), creatine and sodium bicarbonate (n = 1), and sodium bicarbonate and beta-alanine (n = 4) were analyzed. A benefit for cyclists’ athletic performnce was found when consuming a caffeine supplement, and a potential positive effect was noted after the consumption of sodium bicarbonate, as well as after the combination of caffeine and creatine. However, no statistically significant effects were identified for the remaining supplements, whether administered individually or in combination.

## 1. Introduction

Cycling is an endurance sport considered one of the most demanding disciplines in which cyclists must perform under a wide range of exercise intensities that demand different physiological capacities [1]. This is due to the existence of a variety of competitions throughout the racing season. This diversity requires different physiological characteristics to meet the demands of the exercise and thus achieve good sports performance [2]. During cycling, there is a broad spectrum of factors that influence performance, ranging from physiological parameters to psychological, nutritional, or body composition aspects [3].

Given the considerable demands that cycling competitions impose on human physiology, different authors have attempted to determine the energy and nutritional demands of cycling. Scientific reports on energy intake during Grand Tours (Giro d’Italia, Tour de France, and Vuelta a España) have shown high energy intake (5415–7740 kcal/day) and high energy expenditure (6070–7815 kcal/day) during these events [4]. The average daily energy intake reported by cyclists in these races is 84 kcal/kg. Contrary to what might be presumed, cyclists are capable of meeting their energy demands throughout Grand Tours, with their macronutrient intake approaching or exceeding that recommended for ultra-endurance exercise [4,5,6].

In most competitions where narrow margins exist between success and defeat, small factors can become determinants of the outcome of sports competitions. With the aim of enhancing their training capacity and performance, elite athletes frequently incorporate multiple nutritional ergogenic aids.

There is a lack of a universally acknowledged and comprehensive classification system for nutritional supplements; hence, diverse proposals are under consideration. One of the most recognized proposals, known as the “ABCD Classification System”, was published in 2021 by the Australian Institute of Sport (AIS) committee [7]. This system categorizes sports foods and nutritional supplements into four groups based on scientific evidence and other practical considerations determining whether a product is safe, permitted, or effective for improving sports performance. Group A includes supplements with strong scientific evidence for use in specific sports situations through evidence-based protocols and are permitted for use according to established best practice guidelines. Within this group, there are three subgroups: sports foods, medical supplements, and performance supplements, as shown in Table 1. Nutritional ergogenic aids are supplements or ingredients that have the capacity to enhance sports performance. Table 2 outlines the main characteristics of the six nutritional ergogenic aids from Group A of the AIS, detailing their primary food sources, mechanism of action, supplementation protocol, impact on sports performance, and potential adverse effects.

Despite the existence of comprehensive narrative reviews on sports supplementation, systematic evaluations of the effects of combinations of supplements of different natures on exercise performance are limited [8]. To the best of the authors’ knowledge, no prior review has been specifically performed for this aspect within the context of cycling.

In this regard, the objective of this systematic review was to assess the effectiveness of using the highest level of supported nutritional supplements in the context of cycling, as well as their potential synergistic effect. The article specifically focused on Group A of the Australian Institute of Sport (AIS) ABCD classification, particularly on the subgroup of performance nutritional ergogenic aids, which include beta-alanine, sodium bicarbonate, caffeine, creatine, nitrates, and glycerol.

## 2. Materials and Methods

### 2.1. Search Strategy

Two literature searches were conducted on 18 March 2024, using the PubMed (MEDLINE), Scopus, and Web of Science databases. The terms used in the first search were (“caffeine supplement*” OR “creatine supplement*” OR “sodium bicarbonate supplement*” OR “nitrate supplement*” OR “beta-alanine supplement*” OR “glycerol supplement*”) AND “cycling”. The terms used in the second search were ((“caffeine” AND “creatine”) OR (“caffeine” AND “sodium bicarbonate”) OR (“caffeine” AND “nitrate”) OR (“caffeine” AND “beta-alanine”) OR (“caffeine” AND “glycerol”) OR (“creatine” AND “sodium bicarbonate”) OR (“creatine” AND “nitrate”) OR (“creatine” AND “beta-alanine”) OR (“creatine” AND “glycerol”) OR (“sodium bicarbonate” AND “nitrate”) OR (“sodium bicarbonate” AND “beta-alanine”) OR (“sodium bicarbonate” AND “glycerol”) OR (“nitrate” AND “beta-alanine”) OR (“nitrate” AND “glycerol”) OR (“beta-alanine” AND “glycerol”)) AND “cycling” AND “supplement*”.

### 2.2. Eligibility Criteria

The present systematic review was prospectively registered in OSF (Registration Protocol: https://doi.org/10.17605/OSF.IO/6YN53) and conducted following the guidelines outlined in the PRISMA (Preferred Reporting Items for Systematic Reviews and Meta-Analyses) 2020 statement [9]. The inclusion criteria were defined according to the PICOS criteria. Within this framework, the study aimed to enroll healthy adults as the target population. The intervention of interest encompassed the supplementation of caffeine, creatine, sodium bicarbonate, beta-alanine, nitrates, glycerol, or a combination thereof. Placebo was designated as the comparator for eligible studies. The outcomes of interest centered on the measurement of time or power output on a cycle ergometer or bicycle, comprising assessments such as the Wingate test, time trial, or exercise test to exhaustion. Study designs eligible for inclusion involved Randomized Controlled Trials (RCTs) or crossover design studies (CSs).

The inclusion criteria for the first search were studies on humans, conducted on adult populations, published in English, and published between 2019 and 2023. The inclusion criteria for the second search aligned with those established in the first; however, in the second search, the temporal range was expanded to include any studies published from inception to 2023. Literature review studies were excluded.

### 2.3. Study Selection and Data Extraction

Two authors (AVM and PR) independently screened the articles obtained through the search following the established inclusion criteria. Any potential disagreements were resolved by a third reviewer (AL). Initially, duplicate studies were removed. Then, the titles and abstracts of the articles were assessed, followed by a review of the full texts of the studies that met the initial criteria. The entire screening process was performed using Zotero^®^ version 6.0.35.

The data from the included studies were independently extracted by two reviewers (AVM and PR) and thoroughly reviewed by a third reviewer to resolve any potential discrepancies (AL). Data extraction from the selected articles was performed using a standard table designed in Microsoft Word^®^ version 16.83. The following data were collected from the eligible studies: authors and publication date; study design; participant characteristics (sex, age, Body Mass Index (BMI), maximal aerobic capacity (VO_2_ max), and peak maximal capacity (VO_2_ peak)); supplementation protocol (dosage and duration); exercise protocol; and outcomes obtained (magnitude of improvement and level of significance).

### 2.4. Quality Assessment

The methodological quality of the included studies was independently assessed by two reviewers (AVM and PR), with a third reviewer (AL) providing oversight to resolve any potential discrepancies. The “Physiotherapy Evidence Database (PEDro)” scale was used (see Table 3) [10]. A standard table was designed using Microsoft Word^®^ version 16.83. The PEDro scale comprises 11 items for evaluating the methodological quality of clinical trials. Each item is scored as 0 (criterion not met) or 1 (criterion met). The maximum score is 11. Studies scoring from 9 to 11 were considered of high methodological quality, those scoring from 6 to 8 were considered of moderate quality, those scoring from 4 to 5 were rated as low quality, and studies scoring < 3 were considered of very low quality.

## 3. Results

### 3.1. Search and Study Characteristics

A total of 326 results were identified after conducting searches on PubMed (n = 60), Scopus (n = 86), and Web of Science (n = 180). Subsequently, following the exclusion of duplicates (n = 144), the remaining total was 182. After the inclusion criteria were evaluated, 141 articles were excluded based on title and abstract screening, resulting in a total of 41 articles. Additionally, a total of five articles were excluded after full-text screening due to combining supplemental treatment with hypoxia [11], lacking a control group consuming a placebo [12], not measuring performance variables such as time or power output [13], and not meeting the inclusion requirements for study design, as they were not randomized studies [14,15]. Finally, 36 articles were selected for this systematic review (Figure 1).

This section outlines the authors and publication date, study design, participant characteristics, supplementation protocol, exercise protocol, and results obtained. Out of the 36 identified trials, 5 studies examined the effectiveness of caffeine supplementation [16,17,18,19,20]. Two studies focused on analyzing the effects of creatine as a supplement [21,22], while six others focused on sodium bicarbonate [23,24,25,26,27,28]. Beta-alanine supplementation was evaluated in three studies [29,30,31], and eight studies assessed nitrates [32,33,34,35,36,37,38,39]. Notably, none of the trials specifically addressed glycerol supplementation.

Additionally, three studies evaluated the combined supplementation of caffeine and creatine in the context of cycling [40,41,42], while three studies investigated the combined use of caffeine and sodium bicarbonate [43,44,45]. One study examined the concurrent consumption of caffeine and nitrates [46], and another focused on supplementation with creatine and sodium bicarbonate [47]. Furthermore, four studies focused on combined supplementation with sodium bicarbonate and beta-alanine [48,49,50,51].

A total of 701 participants were included in the studies. Participant characteristics varied between 19 and 52 years of age, with BMIs ranging from 20.6 to 29.7 kg/m^2^, the VO_2_ max ranging from 41 to 71.1 mL/kg/min, and the VO_2_ peak ranging from 41.4 to 63.2 mL/kg/min. Regarding the design of the studies included in the systematic review, there were a total of nine Randomized Controlled Trials and 25 crossover studies.

### 3.2. Quality Assessment

In Table 4, an analysis of the methodological quality of the studies included in the systematic review is provided using the “Physiotherapy Evidence Database (PEDro)” scale. The methodological quality of the studies examined in this review ranged from 6 to 10 points, with an average score of 7.75 points. A total of 9 articles exhibited high methodological quality, while 25 articles demonstrated moderate methodological quality. No articles were rated as having low or very low methodological quality.

### 3.3. Caffeine

A total of five studies investigated caffeine supplementation during cycling [16,17,18,19,20]. Four of them utilized the lowest recommended caffeine dosage (3 mg of caffeine per kg of body weight), while the remaining study employed a dosage of 6 mg of caffeine per kg of body weight [17]. Regarding the timing of the supplementation, the majority of the studies recommended consuming caffeine 60 min prior to exercise, while only one suggested doing so 90 min beforehand [16].

The efficacy of caffeine was assessed through time trials of various intensity levels and distances [16,17,20], as well as an adapted Wingate test [19] and an intermittent sprint test [18]. All studies included in the review revealed significant improvements in cycling performance, reflected in a 1.7 to 4.6% improvement in time needed to complete time trials and an increase of 2.53% in average and maximum power output for sprinting. Notably, although three of the five studies included participants of both sexes [16,19,20], no significant differences were observed between them (Table 5).

Among the studies included in this review that focused on investigating the effectiveness of caffeine supplementation in cycling, they demonstrated significant improvements in athletic performance in short-distance, high-intensity, and long-duration tests at moderate intensity.

### 3.4. Creatine

Two studies on creatine supplementation were reviewed [21,22]. Both studies implemented a rapid loading protocol (Table 6).

In the study by Gordon et al. [22], 39 women performed an intermittent sprint test on a cycle ergometer. On the other hand, Schäfer et al. [21] conducted four to five constant load tests (80 rpm) on a cycle ergometer with 11 trained men, measuring the time to failure, which was approximately 3 min in both groups. The first study did not report significant improvements, while the second study observed a prolongation of time to fatigue by 11% in a constant load test.

Thus, the reviewed studies revealed mixed results in short-duration, high-intensity tests. Additionally, it is important to note that only two studies were included in this review, both of which focused on short-duration tests.

### 3.5. Sodium Bicarbonate

Among the six studies included in the review, a dose of 0.3 g of sodium bicarbonate per kilogram of body weight was used in four studies [23,24,25,26], while the lowest dose within the recommended range (0.2–0.4 g of sodium bicarbonate per kg of body weight) was used in two studies [27,28]. The supplementation protocols showed wide variability, ranging from 60 to 180 min prior to exercise (Table 7).

To evaluate the efficacy of sodium bicarbonate, high-intensity tests were conducted, including tests for exhaustion [23], time trials with distances between 2 and 4 km [24,27], and various high-intensity interval exercises [26]. In three out of the six analyzed studies, non-significant results were obtained regarding performance in the tests conducted [24,26,28]. In the study by Thomas et al. [26], which involved eight elite cyclists, although no significant improvements in performance were found, an improvement in perceived effort was observed among cyclists who consumed sodium bicarbonate compared to the placebo group. On the other hand, three studies reported significant improvements, with the magnitude of improvement ranging from 1.6 to 12% [23,25,27].

Regarding sodium bicarbonate supplementation, variable results were observed in its effectiveness for improving cycling performance, with half of the studies showing significant improvements.

### 3.6. Beta-Alanine

The three selected articles for the review implemented the same supplementation protocol, in which 6.4 g of beta-alanine were administered daily over a period of 4 weeks to recreationally trained adult men in endurance sports [29,30,31].

The efficacy of beta-alanine use was evaluated through high-intensity exercises, including tests at 110% of their individual maximal work (W max) [30] and an adapted Wingate test, which involved repeated 10-s sprints and maximal effort tests [29]. Perim et al. [31] opted to implement a simulated road cycling protocol consisting of a 120-min constant load test on a cycle ergometer, with six maximum intensity sprints every 20 min during the test, followed by a 4-km time trial using a road bike [31]. None of the three studies reported statistically significant improvements in any of the performance measures evaluated during cycling.

In summary, the three studies considered in this review on beta-alanine supplementation did not show significant improvements in cycling performance during short-duration, high-intensity exercise or in a simulated road cycling protocol (Table 8).

### 3.7. Nitrates

Eight studies investigating nitrate supplementation during cycling were examined [32,33,34,35,36,37,38,39]. In one study, a dose of 6.4 mmol of nitrates was administered [34], while doses of 8 mmol of nitrates were used in two studies [36,38], a dose of 9.9 mmol of nitrates in one study [35], doses ranging from 12.4 to 13 mmol of nitrates in two studies [33,39], and a dose of 18.5 mmol of nitrates in another study [37]. One study mentioned only the consumption of 340 mg of nitrates without specifying the equivalence in mmol [32]. The supplementation protocols varied between 2 and 3 h prior to exercise, except in the study by Hennis et al. [37], which opted for chronic supplementation over 4 days.

To assess the efficacy of nitrates, various constant load tests were conducted, covering different intensities and durations [35,36,37,38], as well as incremental load tests to exhaustion [39], in addition to tests at 170% of their individual maximal work (W max) [32], a 10-km time trial [33], and a Wingate test consisting of successive 30-s sprints [34].

Significant improvements in cycling performance were identified in two out of the eight studies included in the review [33,34]. The studies that demonstrated these improvements were by Jodra et al. [34], who conducted a Wingate test on a cycle ergometer, and Rokkedal-Lausch et al. [33], who performed a 10-km test on a cycle ergometer. Both observed improvements in the time required to complete the exercise and in the power output during the tests. However, the remaining studies did not find statistically significant improvements in performance.

In summary, the results observed in this review regarding nitrate supplementation have been inconclusive (Table 9). Of the eight included studies, only two reported significant improvements in cycling performance. These studies included short-duration, high-intensity tests.

### 3.8. Glycerol

Three articles were identified as the result of the search. Of these three articles, two were not conducted on humans, and one did not investigate glycerol supplementation in the context of sports. Therefore, no study examining the impact of glycerol supplementation on performance in the context of cycling met the requirements for inclusion in the present review.

### 3.9. Synergies between Supplements

Although both this literature review and the AIS Group A classification address ergogenic aids, our review diverges by concentrating specifically on their efficacy within the context of cycling. This focus facilitates a nuanced examination pertinent to the distinct physiological and performance demands of cycling, assessing the effectiveness of these ergogenic aids in isolation, as well as their potential synergistic effects (Table 10).

The possible synergistic effect of caffeine and creatine supplementation was investigated in three studies [40,41,42]. Two of them, conducted by Lee et al. [41,42], reported significant improvements in cycling performance. An increase in mean power and peak power applied during a high-intensity cycling test on a cycle ergometer was observed, along with a 4.5% improvement in time to fatigue in the group supplemented with caffeine and creatine compared to the group supplemented with creatine and a placebo. However, the study conducted by Vanakoski et al. [40] showed non-significant improvements in variables such as total power applied and maximum pedaling speed during constant-load and moderate-duration cycling tests on a cycle ergometer.

Three articles were included in the review investigating the combined use of caffeine and sodium bicarbonate in cycling [43,44,45]. Among them, only Correia-Oliveira and Lopes-Silva [45] reported improvements in performance in a 4-km time trial. They observe a 2.3% reduction in the time required to complete the exercise and a 4.97% increase in applied power after supplementing with caffeine and sodium bicarbonate compared to the groups that consumed sodium bicarbonate or a placebo. However, in two other studies, no significant improvements in performance were detected when participants performed a 3-km time trial [43] or high-intensity cycling exercises to exhaustion on a cycle ergometer [44].

The only study investigating the combination of caffeine and nitrates in cycling did not find a synergistic effect when performing a 20-km time trial [46], as there were no significant differences between groups (*p* > 0.05).

Morris et al. (2016) did not observe a synergistic effect when investigating the combined supplementation of creatine and sodium bicarbonate, as significant improvements were only obtained in the group supplemented with creatine but not when incorporating sodium bicarbonate [47].

Four studies investigating the combined use of sodium bicarbonate and beta-alanine in cycling were included [48,49,50,51]. None of them found significant improvements in cycling performance when supplemented with sodium bicarbonate and beta-alanine, either in a four-minute test [49] or in high-intensity, short-duration tests [48,50,51], in parameters such as time to exhaustion, time to complete the test, or average power applied during the exercise.

Consequently, there is a lack of consensus in the results of the various studies regarding the different combinations. The combination of supplements that showed the most favorable results in terms of performance enhancement during cycling was caffeine and creatine (two out of three studies).

## 4. Discussion

The present systematic review comprehensively addressed the topic of ergogenic aids in cycling, specifically analyzing the effects of six nutritional supplements—caffeine, creatine, sodium bicarbonate, beta-alanine, nitrates, and glycerol—both individually and in combination.

The studies included in the review have shown favorable results for caffeine regarding the improvement of athletic performance in cyclists, reducing the time needed to complete the exercise by 1.7% to 4.6% [16,17,20] or increasing the force applied during exercise by 2.6% to 4.8% [17,19], both in short-distance and high-intensity tests, as well as in longer-duration tests at moderate intensity. On the other hand, supplementation with sodium bicarbonate has shown a certain tendency to exert a positive effect on athletic performance in cycling, with several studies showing an improvement of between 1.6% and 2.3% in the time needed to complete time trial tests [25,27] and a 12% improvement in time to exhaustion in a constant load test on a cycle ergometer [23]. These findings are consistent with previous evidence in other sports disciplines and support the effectiveness of these supplements in the context of cycling [52,53].

However, limitations and important considerations have been identified for the other supplements evaluated. The results for creatine, nitrates, and beta-alanine have been inconclusive regarding their effectiveness in improving performance in cycling, showing mostly trivial or non-significant improvements, both in short-distance and high-intensity tests, as well as in moderate-intensity and long-duration tests. However, solid conclusions could not be drawn regarding glycerol supplementation due to the lack of available studies addressing this issue. These findings differ from those of previous research in other sports disciplines [54,55,56,57].

Regarding the potential synergistic activity between the various dietary supplements analyzed in this review, only sparse evidence supports a possible beneficial effect derived from the combined supplementation of caffeine and creatine [41,42], showing an improvement of 4.5% in time to exhaustion [42]. For the other combinations of supplements, the existing scientific literature has not yet demonstrated a statistically significant synergistic effect.

## 5. Limitations and Future Research Lines

Some of the limitations of the studies included in this systematic review are related to the wide variability in supplement doses and timings used across different studies, as well as differences in participants’ training levels. These discrepancies hinder direct result comparisons and definitive conclusion extractions.

Additionally, ergogenic aids may occasionally exhibit ergolytic effects. Adverse reactions, such as gastrointestinal distress or other secondary effects, can lead to a worsened performance or prevent subjects from completing trials.

The limited availability of studies on some supplements is a constraint of this systematic review, as conclusions could not be drawn regarding glycerol supplementation or several combinations of supplements in search of a potential ergogenic effect. Moreover, the suitability of the physical tests employed and their representativeness of competitive cycling exertions present additional limitations.

This lack of consensus among studies emphasizes the need for further research to better understand the effects of these nutritional ergogenic aids in the specific discipline of cycling. Given these considerations, future research in the field of ergogenic aids in cycling should include randomized controlled double-blind studies with larger, well-defined samples of cyclists across diverse training levels. Standardized protocols for supplement administration, whether used individually or in combination with others, should be implemented. Additionally, a comprehensive assessment of various performance variables is essential.

## 6. Conclusions

The current systematic review demonstrated the performance benefits of caffeine supplementation for cyclists, a potential positive effect of sodium bicarbonate ingestion, and a combination of caffeine and creatine. The insights garnered from the studies within this systematic review offer practical guidance for sports nutritionists, coaches, and cyclists. These insights provide relevant information regarding the efficacy of various supplementation strategies, thus facilitating the optimization of nutritional planning and sporting performance within the competitive cycling milieu.

## Figures and Tables

**Figure 1 nutrients-16-01768-f001:**
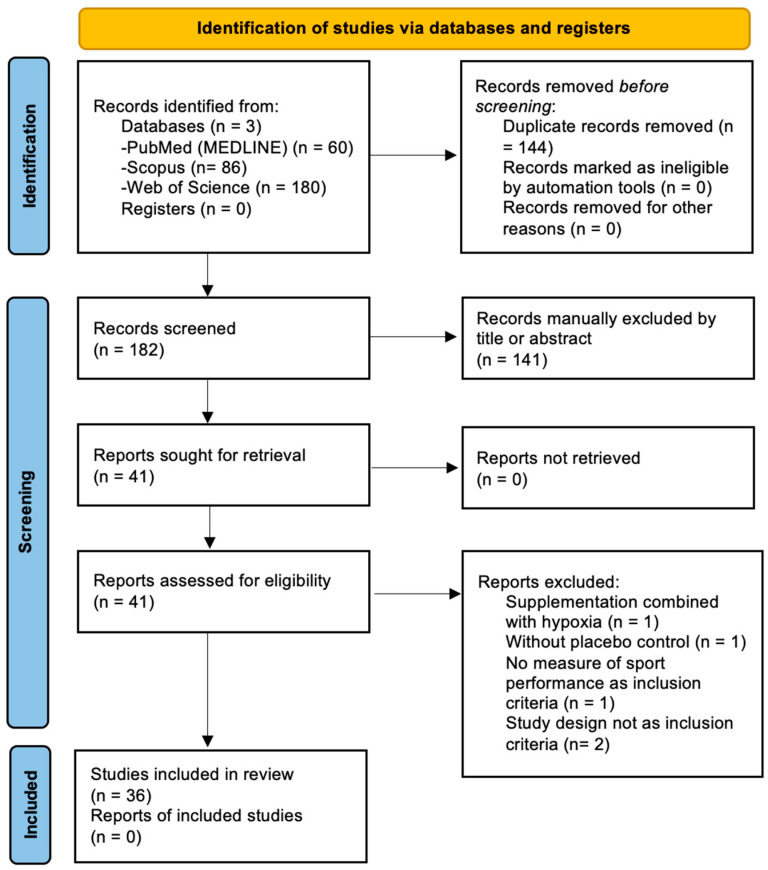
PRISMA flow diagram of the record identification, screening, and selection processes.

**Table 1 nutrients-16-01768-t001:** Group A of the ABCD Classification System of the AIS Sports Supplement Framework: 2021.

Category	Examples
Sports foods	Sports Drink Sports Gels Sports Confectionary Sports Bar Electrolyte supplements Mixed Macronutrient Supplement (Bar, Powder, Liquid Meal)
Medical supplements	Iron Calcium Vitamin D Multivitamin Probiotics Zinc
Performance supplements	Caffeine Beta-alanine Bicarbonate Beetroot juice/Nitrate Creatine Glycerol

Adapted from the Australian Institute of Sport (AIS).

**Table 2 nutrients-16-01768-t002:** Description of the main characteristics of nutritional ergogenic aids from Group A of the ABCD classification by the Australian Institute of Sport (AIS).

Nutritional Ergogenic Aid	Dietary Sources	Mechanism of Action	Supplementation Protocol	Impact on Athletic Performance	Adverse Effects
Caffeine	Coffee, tea, cocoa, energy drinks	Antagonism of adenosine receptors, increased neurotransmitter release, increased availability of myofibrillar calcium, increased mobilization of fatty acids	3–6 mg of caffeine/kg of body weight (BW) 45–60 min before exercise	Improvement of neuromuscular function and skeletal muscle contraction. Reduced depletion of glycogen stores. Enhanced thermoregulatory response. Improved alertness and reaction time. Decreased perception of effort. Enhanced performance in various sports: endurance, high-intensity, team sports, strength-power activities, and submaximal exercises	Gastrointestinal discomfort, insomnia, irritability, tachycardia, arrhythmia
Creatine	Herring, salmon, beef, tuna, pork, cod	Enhancement of intramuscular creatine concentrations, aiding in the maintenance of ATP availability	Loading protocol: 0.3 g of creatine/kg of BW daily for 1 week Slow loading protocol/Maintenance: 0.03 g of creatine/kg of BW daily for 4 weeks	Improvement in performance during high-intensity, short-duration efforts (2–30 s). Increased speed of recovery and delay in the onset of fatigue. Enhanced hydration.	Increase in body weight due to water retention
Sodium Bicarbonate	Water	Increase in extracellular pH buffering capacity	0.2–0.4 g of sodium bicarbonate/kg of BW 60–90 min before exercise	Improvement in performance in short-duration and intermittent exercises (1–7 min). Delay in the onset of muscle fatigue	Gastrointestinal discomfort, diarrhea, nausea
Beta-alanine	Chicken, beef, pork, salmon, turkey, tuna	Increase in muscle carnosine levels, enhancement of intracellular pH buffering capacity	65 mg of beta-alanine per kg of BW every day for 4 weeks.	Improvement in performance in exercises of fixed and intermittent duration (30 s to 10 min). Delay in the onset of muscle fatigue.	Paresthesia in extremities
Nitrates	Lettuce, arugula, spinach, celery, beetroot	Nitric oxide precursors, vasodilation, and increased oxygen transport	310 and 560 mg of nitrates (6–8 mmol) 2–3 h before exercise	Improvement in performance in high-intensity, short-duration exercises (12 to 40 min). Delay in the onset of muscular fatigue in long-duration activities. Improvement in the energy cost of force production. Enhancement of the function of type II muscle fibers.	Gastrointestinal discomfort
Glycerol	Any source of dietary fat and food additive (E-422)	Increased water retention, plasma volume expansion, reduced diuresis	1.2–1.4 g of glycerol per kg of BW dissolved in 25–30 mL of liquid per kg of BW 90–180 min before exercise	Improvement in performance in long-duration sports and in hot or humid environments. Delay in dehydration. Increase in gluconeogenesis. Delay in fatigue perception.	Increase in BW due to water retention. Gastrointestinal discomfort

**Table 3 nutrients-16-01768-t003:** “Physiotherapy Evidence Database (PEDro)” scale criteria.

Criteria		Yes	No
1.	eligibility criteria were specified	1	0
2.	subjects were randomly allocated to groups (in a crossover study, subjects were randomly allocated an order in which treatments were received)	1	0
3.	allocation was concealed	1	0
4.	the groups were similar at baseline regarding the most important prognostic indicators	1	0
5.	there was blinding of all subjects	1	0
6.	there was blinding of all therapists who administered the therapy	1	0
7.	there was blinding of all assessors who measured at least one key outcome	1	0
8.	measures of at least one key outcome were obtained from more than 85% of the subjects initially allocated to groups	1	0
9.	all subjects for whom outcome measures were available received the treatment or control condition as allocated or, where this was not the case, data for at least one key outcome was analyzed by “intention to treat”	1	0
10.	the results of between-group statistical comparisons are reported for at least one key outcome	1	0
11.	the study provides both point measures and measures of variability for at least one key outcome	1	0

Adapted from the Physiotherapy Evidence Database (https://pedro.org.au) (accessed on 5 March 2024).

**Table 4 nutrients-16-01768-t004:** Evaluation of the methodological quality of the studies included in the systematic review (n = 36).

Reference	1	2	3	4	5	6	7	8	9	10	11	Total Score
Vanakoski et al., 1998 [40]	0	1	0	0	1	1	0	1	1	1	1	7/11
Lee et al., 2011 [41]	0	1	0	0	1	1	0	1	1	1	1	7/11
Sale et al., 2011 [48]	0	0	0	1	1	1	0	1	1	1	1	7/11
Lee et al., 2012 [42]	0	1	0	0	1	1	0	1	1	1	1	7/11
Kilding et al., 2012 [43]	0	1	0	0	1	1	1	1	1	1	1	8/11
Bellinger et al., 2012 [49]	0	1	0	0	1	1	0	1	1	1	1	7/11
Danaher et al., 2014 [50]	0	1	0	1	1	1	0	1	1	1	1	8/11
Glaister et al., 2015 [46]	0	1	0	0	1	1	0	1	1	1	1	7/11
Higgins et al., 2016 [44]	0	0	0	0	1	1	0	1	1	1	1	6/11
Morris et al., 2016 [47]	1	1	0	1	1	1	0	1	1	1	1	9/11
Skinner et al., 2019 [16]	1	1	1	1	1	1	0	1	1	1	1	10/11
Schäfer et al., 2019 [21]	0	1	0	1	1	0	0	1	1	1	1	6/11
Da Silva et al., 2019 [51]	1	1	0	1	1	1	0	1	1	1	1	9/11
Wang et al., 2019 [29]	0	1	1	1	1	1	1	0	1	1	1	9/11
Pawlak-Chaouch et al., 2019 [32]	1	1	0	0	1	0	0	1	1	1	1	7/11
Rokkedal-Lausch et al., 2019 [33]	1	1	0	0	1	1	0	1	1	1	1	8/11
Ferreira et al., 2019 [23]	1	1	0	0	1	1	0	0	1	1	1	7/11
Jodra et al., 2019 [34]	1	1	0	0	1	1	0	1	1	1	1	8/11
Nakamura et al., 2020 [18]	0	1	0	0	1	1	0	1	1	1	1	7/11
Hilton et al., 2020 [25]	1	1	0	0	1	1	0	1	1	1	1	8/11
Morales et al., 2020 [17]	1	1	1	1	1	1	0	1	1	1	1	10/11
Voskamp et al., 2020 [24]	1	1	0	0	1	1	0	1	1	1	1	8/11
Berry et al., 2020 [35]	1	1	1	1	1	1	1	0	1	1	1	10/11
Lara et al., 2021 [19]	1	1	1	0	1	1	0	1	1	1	1	9/11
Clarke and Richardson 2021 [20]	1	1	0	0	1	1	0	1	1	1	1	8/11
Thomas et al., 2021 [26]	0	1	0	0	1	1	0	1	1	1	1	7/11
Patel et al., 2021 [30]	0	1	0	1	1	1	1	1	1	1	1	9/11
Thurston et al., 2021 [36]	0	1	0	0	1	0	0	1	1	1	1	6/11
Gough et al., 2022 [27]	0	1	0	0	1	1	1	1	1	1	1	8/11
Perim et al., 2022 [31]	1	1	0	1	1	1	0	0	1	1	1	8/11
Rowland et al., 2022 [38]	0	1	0	1	1	1	0	1	1	1	1	8/11
Hennis et al., 2022 [37]	0	1	0	0	1	1	0	1	1	1	1	7/11
Zhou et al., 2022 [28]	1	1	0	1	0	0	0	0	1	1	1	6/11
Correia-Oliveira and Lopes-Silva 2022 [45]	0	0	0	0	1	1	0	1	1	1	1	6/11
Rowland et al., 2023 [39]	0	1	0	1	1	1	0	1	1	1	1	8/11
Gordon et al., 2023 [22]	1	1	1	1	1	1	0	0	1	1	1	9/11

**Table 5 nutrients-16-01768-t005:** Description of the studies investigating caffeine supplementation in cycling (n = 5).

Reference	Study Design	Participant Characteristics	Supplementation Protocol	Exercise Protocol	Results
Skinner et al., 2019 [16]	CS	27 trained cyclists and triathletes (11 F, 16 M) Age: 29.7 ± 5.3 (F), 32.6 ± 8.3 (M) y/o Body Mass Index (BMI): 21.9 ± 2.7 (F), 24.0 ± 1.3 (M) kg/m^2^ VO_2_ max: 51.9 ± 7.2 (F), 60.4 ± 4.1 (M) mL/kg/min	3 mg caffeine (CAF)/kg of body weight (BW), 90 min before exercise	TT (75% W max) in cycle ergometer	↓ time to complete TT, both in F (*p* = 0.03) and in M (*p* < 0.001), with a similar magnitude (4.3 and 4.6%)
Morales et al., 2020 [17]	CS	14 M recreationally trained cyclists Age: 34.1 ± 4.4 y/o BMI: 24.6 ± 2.1 kg/m^2^ VO_2_ max: 51.5 ± 6.3 mL/kg/min	6 mg CAF/kg of BW, 60 min before exercise	16 km TT (50% W max) in cycle ergometer	↓ time to complete TT (2.63%) and ↑ power output (2.53%) (*p* < 0.05)
Nakamura et al., 2020 [18]	CS	8 M recreationally trained Age: 19.9 ± 0.3 y/o BMI: 21.57 ± 3.61 kg/m^2^ VO_2_ max: 50.0 ± 3.1 mL/kg/min	3 mg CAF/kg of body weight (BW), 60 min before exercise	Intermittent sprint test in cycle ergometer	↑ total work (*p* < 0.05) = mean and peak power output (*p* > 0.05)
Lara et al., 2021 [19]	CS	20 recreationally trained adults (10 F, 10 M) Age: 30.8 ± 5.4 (F), 31.5 ± 7.7 (M) y/o BMI: 21.05 ± 4.18 (F), 22.55 ± 3.45 (M) kg/m^2^ VO_2_ max: 48.1 ± 7.3 (F), 44.7 ± 10.3 (M) mL/kg/min	3 mg CAF/kg of BW, 60 min before exercise	Adapted Wingate test in cycle ergometer	↑ mean and peak power output (*p* < 0.05)
Clarke and Richardson 2021 [20]	CS	46 recreationally trained adults (19 F, 27 M) Age: 28 ± 6 (F), 29 ± 6 (M) y/o BMI: 26.15 ± 6 (F), 24 ± 5 (M) kg/m^2^ VO_2_ max: 41 ± 9 (F), 55 ± 11 (M) mL/kg/min	3 mg CAF/kg of BW, 60 min before exercise	5 km TT in cycle ergometer	↓ time to complete TT (1.7%) (*p* < 0.001)

Participant characteristics data presented as the mean ± standard deviation. Abbreviations: M, male; F, female; y/o, years old; BMI, Body Mass Index; VO_2_ max, Maximal Oxygen Consumption; VO_2_ peak, Peak Oxygen Consumption; RCT, Randomized Controlled Trial; CS, crossover study; CAF, caffeine, CRE, creatine, SB, sodium bicarbonate; NIT, nitrates; BJ, beetroot juice; BA, beta-alanine; BW, body weight; TT, time trial; W max, maximal aerobic power output; VT1, first ventilatory threshold; VT2, secondary ventilatory threshold; s, seconds; min, minutes; h, hours; wks, weeks; ↑, statistically significant increase in the intervention group compared to the placebo group; ↓, statistically significant reduction in the intervention group compared to the placebo group; =, no statistically significant differences between groups.

**Table 6 nutrients-16-01768-t006:** Description of the studies investigating creatine supplementation in cycling (n = 2).

Reference	Study Design	Participant Characteristics	Supplementation Protocol	Exercise Protocol	Results
Schäfer et al., 2019 [21]	RCT	11 M recreationally trained Age: 22.6 ± 2.8 y/o VO_2_ peak: 51.7 ± 8.3 mL/kg/min	20 g creatine (CRE)/day for 5 days	Constant load test to exhaustion in cycle ergometer	↑ time to exhaustion (11%) (*p* = 0.017)
Gordon et al., 2023 [22]	CS	39 F Age: 24.6 ± 5.9 y/o BMI: 21.9 ± 2.7 kg/m^2^	20 g CRE/day for 5 days	Intermittent sprint test in cycle ergometer	= mean power output (*p* > 0.05)

Participant characteristics data presented as the mean ± standard deviation. Abbreviations: M, male; F, female; y/o, years old; BMI, Body Mass Index; VO_2_ max, Maximal Oxygen Consumption; VO_2_ peak, Peak Oxygen Consumption; RCT, Randomized Controlled Trial; CS, crossover study; CAF, caffeine, CRE, creatine, SB, sodium bicarbonate; NIT, nitrates; BJ, beetroot juice; BA, beta-alanine; BW, body weight; TT, time trial; W max, maximal aerobic power output; VT1, first ventilatory threshold; VT2, secondary ventilatory threshold; s, seconds; min, minutes; h, hours; wks, weeks. ↑, statistically significant increase in the intervention group compared to the placebo group; =, no statistically significant differences between groups.

**Table 7 nutrients-16-01768-t007:** Description of the studies investigating sodium bicarbonate supplementation in cycling (n = 6).

Reference	Study Design	Participant Characteristics	Supplementation Protocol	Exercise Protocol	Results
Ferreira et al., 2019 [23]	CS	21 M recreationally trained cyclists Age: 20 ± 2 y/o BMI: 29.7 ± 3.7 kg/m^2^	0.3 g SB/kg of BW, 30 min before exercise	Constant load test to exhaustion in cycle ergometer	↑ time to exhaustion (12%) (*p* = 0.01)
Voskamp et al., 2020 [24]	CS	32 competitive cyclists (16 F, 16 M) Age: 26.3 ± 6 (F), 27.6 ± 6.9 (M) y/o VO_2_ max: 52.3 ± 2.4 (F), 61.8 ± 4.3 (M) mL/kg/min	0.3 g SB/kg of BW, 150 min before exercise	2 km TT in cycle ergometer	= time to complete TT (*p* > 0.05)
Hilton et al., 2020 [25]	CS	11 M cyclists Age: 32 ±12 y/o BMI: 25.2 ± 4.2 kg/m^2^ VO_2_ peak: 63.2 ± 4.9 mL/kg/min	0.3 g SB/kg of BW, 180 min before exercise	4 km TT in cycle ergometer	↓ time to complete TT (2.3%) (*p* = 0.044)
Thomas et al., 2021 [26]	RCT	8 elite cyclists (2 F, 6 M) Age: 21.5 ± 2.1 (F), 19.8 ± 1.5 (M) y/o BMI: 21.2 ± 1.8 (F), 25.8 ± 2.2 (M) kg/m^2^	0.3 g SB/kg of BW, 90 min before exercise	Constant load test (4 × 1000 m) and Intermittent sprint test (3 × 500 m) in cycle ergometer	= time to complete test (*p* > 0.05) ↓ perceive effort (*p* < 0.05)
Gough et al., 2022 [27]	CS	11 M recreationally trained cyclists Age: 28 ± 6 y/o BMI: 24.9 ± 2.5 kg/m^2^	0.2 g SB/kg of BW, 120 min before exercise	4 km TT in cycle ergometer	↓ time to complete TT (1.6%) (*p* < 0.001)
Zhou et al., 2022 [28]	CS	12 M Age: 22.25 ± 0.75 y/o BMI: 23.19 ± 1.6 kg/m^2^	0.2 g SB/kg of BW, 90 min before exercise	Wingate test in cycle ergometer	= mean power output (*p* = 0.587)

Participant characteristics data presented as the mean ± standard deviation. Abbreviations: M, male; F, female; y/o, years old; BMI, Body Mass Index; VO_2_ max, Maximal Oxygen Consumption; VO_2_ peak, Peak Oxygen Consumption; RCT, Randomized Controlled Trial; CS, crossover study; CAF, caffeine, CRE, creatine, SB, sodium bicarbonate; NIT, nitrates; BJ, beetroot juice; BA, beta-alanine; BW, body weight; TT, time trial; W max, maximal aerobic power output; VT1, first ventilatory threshold; VT2, secondary ventilatory threshold; s, seconds; min, minutes; h, hours; wks, weeks; ↑, statistically significant increase in the intervention group compared to the placebo group; ↓, statistically significant reduction in the intervention group compared to the placebo group; =, no statistically significant differences between groups.

**Table 8 nutrients-16-01768-t008:** Description of the studies investigating beta-alanine supplementation in cycling (n = 3).

Reference	Study Design	Participant Characteristics	Supplementation Protocol	Exercise Protocol	Results
Wang et al., 2019 [29]	RCT	38 M recreationally trained Age: 22.5 ± 2.7 y/o BMI: 23.7 ± 1.8 kg/m^2^	6.4 g BA/day for 4 weeks	Adapted Wingate test and 3 min test to exhaustion in cycle ergometer	= time to complete test and mean power output (*p* > 0.05)
Patel et al., 2021 [30]	CS	19 M recreationally trained Age: 21 ± 2 y/o BMI: 23.4 ± 4 kg/m^2^	6.4 g BA/day for 4 weeks	Intermittent sprint test (110% W max) in cycle ergometer	= time to complete test (*p* > 0.05)
Perim et al., 2022 [31]	RCT	17 M recreationally trained cyclists Age: 38 ± 9 y/o BMI: 23 ± 3.5 kg/m^2^ VO_2_ max: 52.4 ± 8.3 mL/kg/min	6.4 g BA/day for 4 weeks	120 min Constant load test and 6 sprint tests in cycle ergometer 4 km TT	= mean power output (*p* > 0.05) and time to complete TT (*p* = 0.43)

Participant characteristics data presented as the mean ± standard deviation. Abbreviations: M, male; F, female; y/o, years old; BMI, Body Mass Index; VO_2_ max, Maximal Oxygen Consumption; VO_2_ peak, Peak Oxygen Consumption; RCT, Randomized Controlled Trial; CS, crossover study; CAF, caffeine, CRE, creatine, SB, sodium bicarbonate; NIT, nitrates; BJ, beetroot juice; BA, beta-alanine; BW, body weight; TT, time trial; W max, maximal aerobic power output; VT1, first ventilatory threshold; VT2, secondary ventilatory threshold; s, seconds; min, minutes; h, hours; wks, weeks. =, no statistically significant differences between groups.

**Table 9 nutrients-16-01768-t009:** Description of the studies investigating nitrate supplementation in cycling (n = 8).

Reference	Study Design	Participant Characteristics	Supplementation Protocol	Exercise Protocol	Results
Pawlak-Chaouch et al., 2019 [32]	CS	11 M elite cyclists Age: 21.7 ± 3.7 y/o BMI: 20.6 ± 2.1 kg/m^2^ VO_2_ max: 71.1 ± 5.2 mL/kg/min	500 mL BJ (340 mg NIT), 2 h before exercise	Intermittent sprint test (170% W max) to exhaustion in cycle ergometer	= number of sets completed (*p* > 0.05)
Rokkedal-Lausch et al., 2019 [33]	CS	12 M cyclists Age: 29.1 ± 7.7 y/o VO_2_ max: 66.4 ± 5.3 mL/kg/min	140 mL BJ (12.4 mmol NIT), 2.75 h before exercise	10 km TT in cycle ergometer	↑ power output (1.6%) (*p* = 0.019) and ↓ time to complete TT (0.6%) (*p* = 0.024)
Jodra et al., 2019 [34]	CS	15 M endurance trained athletes Age: 23 ± 2 y/o BMI: 23.9 ± 2.1 kg/m^2^	70 mL BJ (6.4 mmol NIT), 2.5 h before exercise	Wingate test in cycle ergometer	↑ power output (4.4%) (*p* = 0.039)
Berry et al., 2020 [35]	CS	15 adults (4 F, 11 M) Age: 52 ± 9 (F), 28 ± 4 (M) y/o BMI: 20.7 ± 1.8 (F), 25.3 ± 2.7 (M) kg/m^2^ VO_2_ peak: 51.1 ± 5 (F), 51.9 ± 5.2 (M) mL/kg/min	120 mL BJ (9.9 mmol NIT), 2 h before exercise	Constant load test (75% W max) in cycle ergometer	= time to complete test (*p* = 0.31)
Thurston et al., 2021 [36]	CS	11 M recreationally trained adults Age: 27 ± 5 y/o VO_2_ max: 42 ± 2 mL/kg/min	140 mL BJ (8.2 mmol NIT), 2 h before exercise	Constant load test (80% W max) in cycle ergometer	= time to complete test (*p* = 0.49)
Hennis et al., 2022 [37]	RCT	27 adults (6 F, 21 M) Age: 28.9 ± 5.2 y/o BMI: 23.6 ± 5.9 kg/m^2^ VO_2_ peak: 51.9 ± 9.9 mL/kg/min	BJ (18.5 mmol de NIT) for 4 days	Constant load test in cycle ergometer	= power output (*p* = 0.274)
Rowland et al., 2022 [38]	RCT	9 M recreationally trained Age: 21 ± 1 y/o BMI: 23.7 ± 3.5 kg/m^2^ VO_2_ peak: 49 ± 5.1 mL/kg/min	8.4 g NIT powder (8 mmol NIT), 2 h and 1 h before exercise	2 h moderate intensity test (VT2) and 60 s sprint test	= power output (*p* = 0.61)
Rowland et al., 2023 [39]	CS	12 M Age: 23 ± 4 y/o BMI: 23.4 ± 4.4 kg/m^2^	140 mL BJ (13 mmol NIT), 2.5 h before exercise	Incremental load test to exhaustion in cycle ergometer	= time to complete test (*p* > 0.05)

Participant characteristics data presented as the mean ± standard deviation. Abbreviations: M, male; F, female; y/o, years old; BMI, Body Mass Index; VO_2_ max, Maximal Oxygen Consumption; VO_2_ peak, Peak Oxygen Consumption; RCT, Randomized Controlled Trial; CS, crossover study; CAF, caffeine, CRE, creatine, SB, sodium bicarbonate; NIT, nitrates; BJ, beetroot juice; BA, beta-alanine; BW, body weight; TT, time trial; W max, maximal aerobic power output; VT1, first ventilatory threshold; VT2, secondary ventilatory threshold; s, seconds; min, minutes; h, hours; wks, weeks; ↑, statistically significant increase in the intervention group compared to the placebo group; ↓, statistically significant reduction in the intervention group compared to the placebo group; =, no statistically significant differences between groups.

**Table 10 nutrients-16-01768-t010:** Description of the studies investigating synergies between supplements in cycling (n = 12).

Reference	Study Design	Participant Characteristics	Supplementation Protocol	Exercise Protocol	Results
Vanakoski et al., 1998 [40]	CS	8 national level athletes (2 F, 6 M) Age: 18–29 y/o	7 mg CAF/kg BW, 70 min before exercise + 0.3 g CRE/kg BW for 3 days	1-min sprint test + 45-min Constant load test in cycle ergometer	= power output and maximum pedaling speed (*p* > 0.05)
Lee et al., 2011 [41]	CS	12 M physically active Age: 19 ± 0.6 y/o BMI: 23 ± 1.8 kg/m^2^	6 mg CAF/kg BW, 60 min before exercise + 0.3 g CRE/kg BW for 5 days	Intermittent sprint test in cycle ergometer	↑ mean and peak power output (*p* < 0.05)
Lee et al., 2012 [42]	CS	12 M Age: 20 ± 1.8 y/o BMI: 22.6 ± 1.6 kg/m^2^	6 mg CAF/kg PC, 60 min before exercise + 0.3 g CRE/kg BW for 5 days	Incremental load test to exhaustion in cycle ergometer	↓ time to complete test (4.5%) (*p* < 0.05)
Kilding et al., 2012 [43]	CS	10 M cyclists Age: 24.2 ± 5.4 y/o BMI: 24.7 ± 2.2 kg/m^2^	3 mg CAF/kg BW, 60 min before exercise + 300 mg de SB/kg BW 90 min before exercise	3-km TT in cycle ergometer	= power output in TT (*p* > 0.05)
Higgins et al., 2016 [44]	CS	13 M recreationally trained Age: 21 ± 3 y/o BMI: 24 ± 3 kg/m^2^ VO_2_ max: 46 ± 8 mL/kg/min	5 mg CAF/kg BW, 60 min before exercise + 300 mg SB/kg BW, 60 min before exercise	High-intensity test to exhaustion in cycle ergometer	= time to exhaustion (*p* = 0.75)
Correia-Oliveira and Lopes-Silva 2022 [45]	CS	10 M recreationally trained cyclists Age: 35.2 ± 7.3 y/o BMI: 23.9 ± 2.7 kg/m^2^ VO_2_ peak: 55.7 ± 7.9 mL/kg/min	5 mg CAF/kg BW, 60 min before exercise + 300 mg SB/kg BW, 100 min before exercise	4-km TT in bicycle	↓ time to complete TT (2.3%) (*p* = 0.03) and ↑ power output (4,97%) (*p* = 0.02)
Glaister et al., 2015 [46]	RCT	14 F elite cyclists Age: 31 ± 7 y/o BMI: 21.6 ± 1.7 kg/m^2^	5 mg CAF/kg BW, 60 min before exercise + 70 mL BJ (7.3 mmol NIT), 180 min before exercise	20-km TT in cycle ergometer	= power output and perceive effort (*p* > 0.05)
Morris et al., 2016 [47]	RCT	63 adults (35 F, 28 M) Age: 22.2 ± 2 y/o BMI: 23 ± 3 kg/m^2^ VO_2_ peak: 41.4 ± 7.6 mL/kg/min	3 g CRE + 1 g SB/day for 8 weeks	Sprint test + 5-km TT in cycle ergometer	= power output and time to complete TT (*p* > 0.05)
Sale et al., 2011 [48]	CS	20 M Age: 25 ± 5 y/o BMI: 25 ± 2.86 kg/m^2^	300 mg SB/kg BW, 180 min before exercise + 6.4 g BA/day for 4 weeks	Constant load test (110% W max) in cycle ergometer	= time to exhaustion (*p* = 0.74)
Bellinger et al., 2012 [49]	CS	14 M cyclists Age: 25.4 ± 7.2 y/o VO_2_ max: 66.6 ± 5.7 mL/kg/min	300 mg SB/kg BW, 90 min before exercise + 65 mg BA/kg/day for 4 weeks	4-min test in cycle ergometer	= mean power output (*p* > 0.05)
Danaher et al., 2014 [50]	CS	8 M recreationally trained Age: 26.2 ± 1.9 y/o BMI: 24.9 ± 0.8 kg/m^2^ VO_2_ peak: 51 ± 2.5 mL/kg/min	300 mg SB/kg BW, 90 min before exercise + 4.8 g BA/day for 4 weeks, and 6.4 g BA/day for 2 weeks	Sprint test + Constant load test (110% W max) in cycle ergometer	= time to exhaustion in test (*p* > 0.05)
Da silva et al., 2018 [51]	RCT	71 M cyclists Age: 37.5 ± 6.8 y/o BMI: 23.55 ± 1.6 kg/m^2^ VO_2_ peak: 60.2 ± 4.5 mL/kg/min	300 mg SB/kg BW 60 min before exercise + 6.4 g BA/day for 4 weeks	Intermittent sprint test (110% W max) + TT in cycle ergometer	= time to complete test (*p* = 0.06)

Participant characteristics data presented as the mean ± standard deviation. Abbreviations: M, male; F, female; y/o, years old; BMI, Body Mass Index; VO_2_ max, Maximal Oxygen Consumption; VO_2_ peak, Peak Oxygen Consumption; RCT, Randomized Controlled Trial; CS, crossover study; CAF, caffeine, CRE, creatine, SB, sodium bicarbonate; NIT, nitrates; BJ, beetroot juice; BA, beta-alanine; BW, body weight; TT, time trial; W max, maximal aerobic power output; VT1, first ventilatory threshold; VT2, secondary ventilatory threshold; s, seconds; min, minutes; h, hours; wks, weeks; ↑, statistically significant increase in the intervention group compared to the placebo group; ↓, statistically significant reduction in the intervention group compared to the placebo group; =, no statistically significant differences between groups.

## Data Availability

The data used during the current study are available from the corresponding author upon reasonable request.

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
