# Peer review of "Nutritional Ergogenic Aids in Cycling: A Systematic Review"

_nutrients, 2024, doi:10.3390/nu16111768_

Round 1
Reviewer 1 Report
Comments and Suggestions for Authors
Reviewer Comments to Author
In this study the authors perform a systematic review on nutritional ergogenic aids in cycling. The review is well written and follows all the critical steps in carrying out a systematic review. I have only a few specific points.
Specific points…
L40 – delete ‘considerably’
L49 – delete ‘simultaneously into their diet’ as there are a number of means by which they may be taken
L69 - it is only here that the reader hears how this review adds to the AIS classifications. Consider making it clear within the title.
Table 4 – This table is a lot for a reader to digest. Consider breaking up the studies into the subheadings mentioned in L 154-164. This can lead the reader to studies of interest, and within text.
L173 – VO2 is subscripted in some parts of the table and text, but not other places. Subscript (VO2) throughout.
L267 – change found to find.
L272 – one could refute that a 10 km TT is considered ‘longer duration, moderate intensity’ as it is all out for ~<15 mins.
L273 – consider rewording paragraph as such….
“Three articles were identified as the result of the search. Of these three articles, two were not conducted in humans, and one did not investigate glycerol supplementation in the context of sports. Therefore, no study examining the impact of glycerol supplementation on performance in the context of cycling met the requirements for inclusion in the present review. “
L282 – consider adding a segue sentence or two indicating how this lit review may differ from that of the AIS class A classification. Since both are similar reviews of these ergogenic aids, yet yours is cycling focused.
L287 – reword to be more clear what 4.5% improvement refers too. CAF and creatine compared to CAF alone? Or compared to placebo. This is important for all of the synergistic effects outlined.
L294 – again, these improvements need to be put in perspective.
L315 – somewhere, perhaps here, or later in the limitations section, it should be noted evidence of ergolytic effects. Meaning any evidence that adverse effects outlined in table 2 worsened performance by any of these dietary supplements. This could be outlined in each of the sections, or combined at the end. Often during these studies, authors note that only X number of subjects were able to complete all of the trials due to GI distress (or likewise). Or, if they were able to complete trials, then a worse performance than placebo. Even with no evidence of ergolytic effects, it can be important for the reader to understand this.
Author Response
We would like to thank the reviewers for their insightful comments and suggestions. We have attempted to address all their concerns in this revised paper. All corrections and additions were highlighted in yellow in the manuscript. Please find below point by point answers to your questions.
With respect to the comments from the Reviewer 1:
Comment 1. In this study the authors perform a systematic review on nutritional ergogenic aids in cycling. The review is well written and follows all the critical steps in carrying out a systematic review. I have only a few specific points.
Response: The authors would like to sincerely thank the reviewer for his kind and constructive comments.
Comment 2. L40 – delete ‘considerably’.
Response: Thank you for your comment. According to the suggestion from the Reviewer, “considerable” was deleted.
Comment 3. L49 – delete ‘simultaneously into their diet’ as there are a number of means by which they may be taken.
Response: Thank you for your comment. According to the suggestion from the Reviewer, “simultaneously into their diet’” was deleted.
Comment 4. L69 - it is only here that the reader hears how this review adds to the AIS classifications. Consider making it clear within the title.
Response: Including AIS in the tittle would perhaps make the article less attractive to the reader. According to your suggestion we have included the term “AIs” in the abstract and keywords. We believe that with them it is already visible enough
Comment 5. Table 4 – This table is a lot for a reader to digest. Consider breaking up the studies into the subheadings mentioned in L 154-164. This can lead the reader to studies of interest, and within text.
Response: Thank you for your comment. According to the suggestion from the Reviewer, the Table 4 were spitted into 5 different tables in the revised version of the manuscript, depending on the type of ergonomic aid.
Comment 6. L173 – VO2 is subscripted in some parts of the table and text, but not other places. Subscript (VO2) throughout.
Response: Thank you for your comment. According to the suggestion from the Reviewer, VO2 is subscripted throughout the manuscript.
Comment 7. L267 – change found to find.
Response: Thank you for your comment. According to the suggestion from the Reviewer, “found” was changed to “find”.
Comment 8. L272 – one could refute that a 10 km TT is considered ‘longer duration, moderate intensity’ as it is all out for ~<15 mins.
Response: In the revised version of the manuscript, it was corrected it and put only short duration and high intensity studies, accepting the reviewer's suggestion. Since it is true that being a time trial the cyclists are going to ride at very high intensity and time is limited.
Comment 9: L273 – consider rewording paragraph as such….
“Three articles were identified as the result of the search. Of these three articles, two were not conducted in humans, and one did not investigate glycerol supplementation in the context of sports. Therefore, no study examining the impact of glycerol supplementation on performance in the context of cycling met the requirements for inclusion in the present review. “
Response: Thank you for your comment. The paragraph was rephrased according to the suggestions from the Reviewer.
Comment 10: L282 – consider adding a segue sentence or two indicating how this lit review may differ from that of the AIS class A classification. Since both are similar reviews of these ergogenic aids, yet yours is cycling focused.
Response: Thank you for your comment. According to the suggestion from the Reviewer, it was added the paragraph: “Although both this literature review and the AIS Group A classification address ergogenic aids, our review diverges by concentrating specifically on their efficacy within the context of cycling. This focus facilitates a nuanced examination pertinent to the distinct physiological and performance demands of cycling, assessing the effectiveness of these ergogenic aids in isolation as well as their potential synergistic effects.”
Comment 11: L287 – reword to be more clear what 4.5% improvement refers too. CAF and creatine compared to CAF alone? Or compared to placebo. This is important for all of the synergistic effects outlined.
Response: Thank you for your comment. In the revised version of the manuscript, it was rephrased this paragraph to: “An increase in mean power and peak power applied during a high-intensity cycling test on a cycle ergometer was observed, along with a 4.5% improvement in time to fatigue in the group supplemented with caffeine and creatine compared to the group supplemented with creatine and placebo.”
Comment 12: L294 – again, these improvements need to be put in perspective.
Response: In the same, way, it was added the following paragraph in the revised version of the manuscript: “They observe a 2.3 % reduction in the time required to complete the exercise and a 4.97 % increase in applied power after supplementing with caffeine and sodium bicarbonate, compared to the groups that consumed sodium bicarbonate or placebo.”
Comment 13: L315 – somewhere, perhaps here, or later in the limitations section, it should be noted evidence of ergolytic effects. Meaning any evidence that adverse effects outlined in table 2 worsened performance by any of these dietary supplements. This could be outlined in each of the sections, or combined at the end. Often during these studies, authors note that only X number of subjects were able to complete all of the trials due to GI distress (or likewise). Or, if they were able to complete trials, then a worse performance than placebo. Even with no evidence of ergolytic effects, it can be important for the reader to understand this.
Response: Thank your for your comment. Accordingly, the limitations section was rephrased to the following content: “Some of the limitations of the studies included in this systematic review are related to the wide variability in supplement doses and timing used across different studies, as well as differences in participants' training levels. These discrepancies hinder direct result comparison and definitive conclusion extraction.
Additionally, ergogenic aids may occasionally exhibit ergolytic effects. Adverse reactions, such as gastrointestinal distress or other secondary effects, can lead to worsened performance or prevent subjects from completing trials.
The limited availability of studies on some supplements is a constraint of this systematic review, as conclusions could not be drawn regarding glycerol supplementation or several combinations of supplements in search of a potential ergogenic effect. Moreover, the suitability of the physical tests employed, and their representativeness of competitive cycling exertions present additional limitations.
This lack of consensus among studies emphasizes the need for further research to better understand the effects of these nutritional ergogenic aids in the specific discipline of cycling. Given these considerations, future research in the field of ergogenic aids in cycling should include randomized controlled double-blind studies with larger, well-defined samples of cyclists across diverse training levels. Standardized protocols for supplement administration, whether used individually or in combination with others, should be implemented. Additionally, a comprehensive assessment of various performance variables is essential.”
Reviewer 2 Report
Comments and Suggestions for Authors
The authors have prepared a well written and interesting manuscript. They have detailed well the available research regarding the use of performance supplements for cyclists.
This research will be of interest to professionals who work with cyclists as well as nutrition professionals.
The SR also illustrates current gaps in knowledge
Excellent use of tables and the use of the PEDro
Only feedback: you sometimes include p-values in the tables, and other times you do not. Is there a reason? For example, Clarke 2021 shows a decrease in time to compete; however, was this significant? I would recommend including the p-value for all results
Author Response
Reviewer 2:
Comment 1. The authors have prepared a well written and interesting manuscript. They have detailed well the available research regarding the use of performance supplements for cyclists.
This research will be of interest to professionals who work with cyclists as well as nutrition professionals.
The SR also illustrates current gaps in knowledge
Excellent use of tables and the use of the PEDro
Response: The authors would like to sincerely thank the reviewer for his kind and constructive comments.
Comment 2. Only feedback: you sometimes include p-values in the tables, and other times you do not. Is there a reason? For example, Clarke 2021 shows a decrease in time to compete; however, was this significant? I would recommend including the p-value for all results.
Response: In the original version, only work for which statistically significant differences were found, the p-value was mentioned (perhaps with some omission due to error). In the corrected version of the manuscript, all cited articles p value have been added to the Tables.